# SmoothMix: Training Confidence-calibrated Smoothed Classifiers for Certified Adversarial Robustness

Jongheon Jeong [1]   Sejun Park [2] [*]   Minkyu Kim [3]   Heung-Chang Lee [4]   Doguk Kim [4]   Jinwoo Shin [3] [1]

## Abstract

*Randomized smoothing* is currently a state-of-the-art method to construct a *certifiably robust* classifier from neural networks against $\ell_2$-adversarial perturbations. Under the paradigm, the robustness of a classifier is aligned with the *prediction confidence*, *i.e.*, the higher confidence from a smoothed classifier implies the better robustness. This motivates us to rethink the fundamental trade-off between accuracy and robustness in terms of *calibrating* confidences of smoothed classifier. In this paper, we propose a simple training scheme, coined *SmoothMix*, to control the robustness of smoothed classifiers via *self-mixup*: it trains convex combinations of samples along the direction of adversarial perturbation. The proposed procedure effectively identifies over-confident, near off-class samples as a cause of limited robustness in case of smoothed classifiers, and offers an intuitive way to adaptively set a new decision boundary between these samples for better robustness. Our experiments show that the proposed method can significantly improve the certified $\ell_2$-robustness of smoothed classifiers compared to state-of-the-art robust training methods.

## 1. Introduction

*Adversarial examples* (Szegedy et al., 2014; Goodfellow et al., 2015) in deep neural networks clearly highlight that neural networks often generalize differently from humans, at least without an additional prior of *local smoothness* of predictions with respect to the input space: an adversarially-crafted, yet imperceptible input perturbation can drastically change the prediction of a neural network based classifier.

*Randomized smoothing* (Lecuyer et al., 2019; Cohen et al., 2019) is relatively a recent idea that aims to *indirectly* encode the smoothness prior: Cohen et al. (2019) have shown that any classifier, regardless of whether it is smooth or not, can be transformed into a *certifiably robust* classifier via averaging its predictions over Gaussian noise. Compared to *adversarial training* (Madry et al., 2018) which directly encodes the smoothness by augmenting training data with its adversarial examples, this notion of "indirect" smoothness can be favorable in a sense that (a) it is easier to optimize, and (b) offers a provable guarantee on the robustness.

**Contribution.** In this paper, we propose *SmoothMix*, a novel adversarial training method designed for improving the certified robustness of smoothed classifiers. One of the key features that smoothed classifiers offer is a direct correspondence from *prediction confidence* to adversarial robustness: achieving a higher confidence in a smoothed classifier implies that the classifier can give a better certified robustness. Inspired by this, we found that the certified robustness of a given data sample can be significantly decreased by nearby *off-class* but *over-confident* (Pereyra et al., 2017) inputs: such "harmful" inputs would occupy an unnecessarily large robust radius near the sample of our interest.

Under the finding, we aim to calibrate the confidence of these off-class inputs to improve the certified robustness at the original input. More specifically, we first observe that such over-confident examples can be efficiently found along the direction of *adversarial* perturbations for a given input. Then, we suggest to regularize the over-confident predictions along the adversarial direction toward the *uniform* prediction through a *mixup* loss (Zhang et al., 2018) (see Figure 1 for an overview). This new approach of incorporating adversarial examples effectively permits more distant examples in training, even when they goes off-class, based on the local-smoothness of smoothed classifiers.

Overall, our work suggests that the robustness of a classifier should be set individually per sample considering its nearby inputs: we approach this problem with the relationship between the confidence and robustness of smoothed classifiers. Recently, there have been also some initial attempts to incorporate a *sample-wise* treatment for robustness by allowing input-dependent noise scales in randomized smoothing (Al-

---
[*]Work done at KAIST [1]School of Electrical Engineering, KAIST, Daejeon, South Korea [2]Vector Institute for Artificial Intelligence, Toronto, Canada [3]Graduate School of AI, KAIST, Daejeon, South Korea [4]Kakao Enterprise, Seongnam, Gyeonggi-do, South Korea. Correspondence to: Jongheon Jeong <jongheonj@kaist.ac.kr>, Jinwoo Shin <jinwoos@kaist.ac.kr>.

*Accepted by the ICML 2021 workshop on A Blessing in Disguise: The Prospects and Perils of Adversarial Machine Learning.* Copyright 2021 by the author(s).

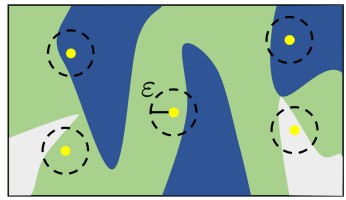

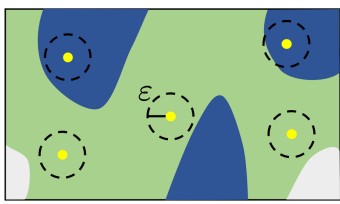

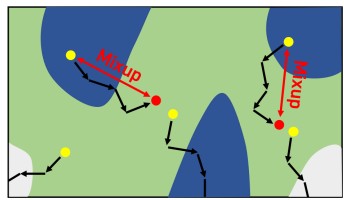

(a) Adversarial training        (b) SmoothAdv        (c) SmoothMix (Ours)

*Figure 1.* Illustrations of how each training method obtains adversarial robustness: adversarial training (Madry et al., 2018) considers an $\varepsilon$-ball around each sample and corrects adversarial examples found in these balls; SmoothAdv (Salman et al., 2019) directly employs adversarial training on smoothed classifiers; SmoothMix (ours) can be differentiated from SmoothAdv as it (*i*) does not assume an explicit norm restriction on adversarial examples, and (*ii*) applies the *mixup* (Zhang et al., 2018) instead of correcting the adversarial examples.

farra et al., 2020; Wang et al., 2021; Chen et al., 2021). However, our theoretical analysis shows that such an approach would eventually suffer from the curse of dimensionality (Theorem 1 in Appendix I), highlighting our approach of focusing on a "better calibration" as a promising alternative.

## 2. Preliminaries

We assume an *i.i.d.* dataset $\mathcal{D} = \{(x_i, y_i)\}_{i=1}^{n} \sim P$, where $x_i \in \mathbb{R}^d$ and $y_i \in \mathcal{Y} := \{1, \cdots, C\}$, and focus on the problem of correctly classifying a given input $x$ into one of $C$ classes. Let $f : \mathbb{R}^d \to \mathcal{Y}$ be a classifier modeled by $f(x) := \arg\max_{c \in \mathcal{Y}} F_c(x)$ with $F : \mathbb{R}^d \to \Delta^{C-1}$, where $\Delta^{C-1}$ denotes the probability simplex in $\mathbb{R}^C$. For example, $F$ can be a neural network followed by a softmax layer. In the context of *adversarial robustness*, one may additionally concern to maximize the *minimum-distance* of adversarial perturbation (Moosavi-Dezfooli et al., 2016; Carlini & Wagner, 2017; Carlini et al., 2019), namely:

$$R(f; x, y) := \min_{f(x') \neq y} \|x' - x\|_2. \tag{1}$$

**Randomized smoothing.** In cases when $f$ is too complex to control its predictions in practice, *e.g.*, if $f$ is a neural network on high-dimensional data, directly solving and maximizing (1) can be hard. *Randomized smoothing* (Cohen et al., 2019) instead construct a new classifier $\hat{f}$ from $f$ that is easier to obtain robustness by transforming the base classifier $f$ with a certain *smoothing measure*, where in this paper we focus on the case of Gaussian distributions $\mathcal{N}(0, \sigma^2 I)$:

$$\hat{f}(x) := \arg\max_{c \in \mathcal{Y}} \mathbb{P}_{\delta \sim \mathcal{N}(0, \sigma^2 I)} (f(x + \delta) = c). \tag{2}$$

For a given $(x, y)$, $R(\hat{f}; x, y)$ can be lower-bounded by the *certified radius* $\underline{R}(\hat{f}, x, y)$, which can be derived from the *confidence* of $\hat{f}$ at $x$, namely we denote it by $p_f(x)$:

$$R(\hat{f}; x, y) \geq \sigma \cdot \Phi^{-1}(p_f(x)) =: \underline{R}(\hat{f}, x, y), \tag{3}$$

$$\text{where } p_f(x) := \mathbb{P}_{\delta \sim \mathcal{N}(0, \sigma^2 I)}(f(x + \delta) = \hat{f}(x)), \tag{4}$$

provided that $\hat{f}(x) = y$, and otherwise $R(\hat{f}; x, y) := 0$. Here, $\Phi$ denotes the cumulative distribution function of the standard normal distribution.

Although randomized smoothing can be applied for any classifier $f : \mathbb{R}^d \to \mathcal{Y}$, the robustness of smoothed classifiers can vary depending on $p_f$ as in (3), *i.e.*, how $f$ performs on a given input under the presence of Gaussian noise. In this sense, to obtain a robust $\hat{f}$, Cohen et al. (2019) simply propose to train $f$ using Gaussian augmentation by default:

$$\min_{F} \mathbb{E}_{\substack{(x,y) \sim P \\ \delta \sim \mathcal{N}(0, \sigma^2 I)}} [\mathcal{L}(F(x + \delta), y)], \tag{5}$$

where $\mathcal{L}$ denotes the standard cross-entropy loss.

## 3. Method

Our goal in this paper is to develop a more suitable form of adversarial training (AT) for smoothed classifiers, taking into account their unique characteristics on adversarial robustness over standard neural networks. Figure 1 illustrates a motivating example: as shown in Figure 1(a), AT typically assumes a fixed-sized ball of radius $\varepsilon$ that each adversarial perturbation must be in, as the goal of the training is to defend the classifier against adversaries under a specific threat model. However, in a case when AT is applied to a smoothed classifier, *e.g.*, as done by *SmoothAdv* (Salman et al., 2019), this assumption may be too restrictive, particularly for inputs where the classifier already certifies robustness of radii larger than $\varepsilon$ (*e.g.*, Figure 1(b)). This demands for a new form of AT specially for smoothed classifiers, *e.g.*, that allows more distant adversarial examples, despite its fundamental difficulty in the context of standard neural networks (Kang et al., 2020; Zhang et al., 2020b).

### 3.1. Exploring over-confident adversarial examples in smoothed classifiers

Recall that we have a (base) classifier $f$ of the form $f(x) = \arg\max_{c \in \mathcal{Y}} F_c(x)$, $\hat{f}$ is its smoothed counterpart, and we aim to improve the robustness of $\hat{f}$ by incorporating adversarial examples in training. In this paper, we are particularly interested in adversarial examples of $\hat{f}$ that is found *without* a hard restriction in its perturbation size. More concretely, for a given training sample $(x, y) \sim P$, we find adversarial

examples by solving the following optimization:

$$\tilde{x} := \arg\max_{x'} \left( \mathcal{L}(\hat{f}; x', y) - \beta \cdot \|x' - x\|_2^2 \right), \quad (6)$$

where $\mathcal{L}$ is the cross-entropy loss, and $\beta > 0$ is to ensure that (6) cannot be arbitrarily far from $x$.

As proposed by Salman et al. (2019), one can optimize (6) by approximating the intractable $\hat{f}$ with the soft-smoothed classifier $\hat{F} := \mathbb{E}_\delta[F(x + \delta)]$: based on this approximation, we simply perform a $T$-step gradient ascent from $\tilde{x}^{(0)} := x$ with step size $\alpha > 0$ to solve (6) using $m$ samples of $\delta$, namely $\delta_1, \cdots, \delta_m \sim \mathcal{N}(0, \sigma^2 I)$:[1]

$$\tilde{x}^{(t+1)} := \tilde{x}^{(t)} + \alpha \cdot \frac{\nabla_x J(\tilde{x}^{(t)})}{\|\nabla_x J(\tilde{x}^{(t)})\|_2}, \quad (7)$$

$$\text{where} \quad J(x) := -\log\left( \frac{1}{m} \sum_i F_y(x + \delta_i) \right). \quad (8)$$

Figure 2 demonstrates two particular instances of these "unrestricted" adversarial examples found from (7) on $x$, and plots how the confidence of inputs changes as they are linearly interpolated from the clean input to its adversarial counterpart $\tilde{x}$. From this, we make several remarks those would lead to a more direct motivation to our method:

- We observe $\tilde{x}$ via (7), *i.e.*, from a smoothed classifier, could contain enough amount of semantic changes even in a perceptual sense, in either ways of translating the input to another class (Figure 2(a)), or simply removing some relevant information for the current class (Figure 2(b)). At least for these cases, therefore, it is reasonable for the classifier to keep their low confidence to the original class. In this sense, we leverage the provable robustness of smoothed classifiers during training to reasonably obtain a semantically off-class samples those to be labeled as the uniform confidence (Santurkar et al., 2019; Kaur et al., 2019).

- A major problem we rather highlight here is the tendency of *over-confidence* (Pereyra et al., 2017) toward the adversarial direction: the adversarial example $\tilde{x}$ usually attain significantly higher confidence compared to those of $x$, consequently their *certified radius* (3) would be much larger as well. Therefore, considering that $\tilde{x}$ are still nearby $x$, such the over-confidence at $\tilde{x}$ would negatively affect the certified radius of $x$, especially when $\tilde{x}$ does not contain much semantically meaningful information as observed in Figure 2(b).

---

[1] Here, we note that the $\beta$-term in (6) are omitted in (7). In practice, we do not use nor tune $\beta$ in our method for simplicity, as the role of $\beta$ can be replaced by assuming a finite $\alpha \cdot T$, *i.e.*, by the *Lagrangian duality*: an unconstrained optimization with $\ell_2$-regularization implicitly defines a hard constraint in its $\ell_2$-norm.

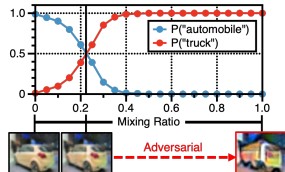 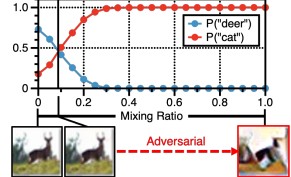

(a) In-class translation          (b) Out-of-class translation

*Figure 2.* Illustration of adversarial examples unrestrictively found in CIFAR-10 with a smoothed ResNet-110 ($\sigma = 0.25$). The plot demonstrates the change of confidence between two classes as the input is linearly interpolated.

### 3.2. SmoothMix for confidence-calibrated training of smoothed classifiers

Based on the observations from Section 3.1, we hypothesize that the *miscalibration* of confidences between $x$ and its unrestricted adversarial example $\tilde{x}$ is an important factor that degrades the certified robustness of smoothed classifiers, and propose to penalize the over-confidence by mixing the *uniform* confidence to them. More concretely, we consider the *mixup* (Zhang et al., 2018) training between $x$ and $\tilde{x}$, *i.e.*, by augmenting the given data with the following pairs:

$$x^{\text{mix}} := (1 - \lambda) \cdot x + \lambda \cdot \tilde{x}^{(T)}, \quad (9)$$

$$y^{\text{mix}} := (1 - \lambda) \cdot \hat{F}(x) + \lambda \cdot \frac{1}{C}, \quad (10)$$

where $\lambda \sim \mathcal{U}\left(\left[0, \frac{1}{2}\right]\right)$ with the uniform distribution $\mathcal{U}$, $\hat{F}(x) \in \Delta^{C-1}$ is the soft-smoothed prediction of $x$, and $\mathbb{1}$ denotes the $C$-dimensional vector of ones. Here, we notice that $\lambda$ is sampled only from $[0, \frac{1}{2}]$, unlike the standard choice (Zhang et al., 2018) of $\mathcal{U}([0, 1])$: recall from Figure 2(a) that $\tilde{x}$ can be often semantically in-class, so that a direct supervision of the uniform confidence on it could harm the classifier. By simply taking only the half part of the mixed samples closer to $x$, we could reasonably avoid these cases while maintaining its effect to prevent the over-confidence issue. The actual loss to minimize for these new data simply follows the cross-entropy loss with Gaussian augmentation, similarly to (5):

$$L^{\text{mix}} := \mathbb{E}_{\delta \sim \mathcal{N}(0, \sigma^2 I)} \left[ \mathcal{L}(F(x^{\text{mix}} + \delta), y^{\text{mix}}) \right]. \quad (11)$$

**Incorporating SmoothAdv for free.** As we focus on adversarial examples that are moderately far from the original inputs assuming that the classifier is already locally-smooth, one may still enjoy the effectiveness of SmoothAdv if it could further enforce the local smoothness. We found that simply taking $x \leftarrow \tilde{x}^{(1)}$ without modifying our current training, *i.e.*, using the *single-step adversarial example* found during (7) instead of the clean sample, can reasonably bring this effect. In this respect, we allow SmoothMix to use $(\tilde{x}^{(1)}, y)$ instead of $(x, y)$ depending on demand of more robustness at expense of decreased clean accuracy.

*Table 1.* Comparison of approximate certified test accuracy (%) and ACR on CIFAR-10. We set our results bold-faced whenever the value improves the Gaussian baseline, and underlined whenever the value improves the best among the considered baselines. * indicates that the results are evaluated from the official pre-trained models released by authors.

| $\sigma$ | Models (CIFAR-10) | ACR | 0.00 | 0.25 | 0.50 | 0.75 | 1.00 | 1.25 | 1.50 | 1.75 |
|---|---|---|---|---|---|---|---|---|---|---|
| 0.25 | Gaussian (Cohen et al., 2019) | 0.424 | 76.6 | 61.2 | 42.2 | 25.1 | 0.0 | 0.0 | 0.0 | 0.0 |
| | Stability training (Li et al., 2019) | 0.421 | 72.3 | 58.0 | 43.3 | 27.3 | 0.0 | 0.0 | 0.0 | 0.0 |
| | SmoothAdv* (Salman et al., 2019) | 0.544 | 73.4 | 65.6 | 57.0 | 47.5 | 0.0 | 0.0 | 0.0 | 0.0 |
| | MACER* (Zhai et al., 2020) | 0.531 | 79.5 | 69.0 | 55.8 | 40.6 | 0.0 | 0.0 | 0.0 | 0.0 |
| | Consistency (Jeong & Shin, 2020) | 0.552 | 75.8 | 67.6 | 58.1 | 46.7 | 0.0 | 0.0 | 0.0 | 0.0 |
| | **SmoothMix (Ours)** | **0.553** | **77.1** | **67.9** | **57.9** | 46.7 | 0.0 | 0.0 | 0.0 | 0.0 |
| | **+ One-step adversary** | **0.548** | 74.2 | **66.1** | **57.4** | **47.7** | 0.0 | 0.0 | 0.0 | 0.0 |
| 0.50 | Gaussian (Cohen et al., 2019) | 0.525 | 65.7 | 54.9 | 42.8 | 32.5 | 22.0 | 14.1 | 8.3 | 3.9 |
| | Stability training (Li et al., 2019) | 0.521 | 60.6 | 51.5 | 41.4 | 32.5 | 23.9 | 15.3 | 9.6 | 5.0 |
| | SmoothAdv* (Salman et al., 2019) | 0.684 | 65.3 | 57.8 | 49.9 | 41.7 | 33.7 | 26.0 | 19.5 | 12.9 |
| | MACER* (Zhai et al., 2020) | 0.691 | 64.2 | 57.5 | 49.9 | 42.3 | 34.8 | 27.6 | 20.2 | 12.6 |
| | Consistency (Jeong & Shin, 2020) | 0.720 | 64.3 | 57.5 | 50.6 | 43.2 | 36.2 | 29.5 | 22.8 | 16.1 |
| | **SmoothMix (Ours)** | **0.715** | 65.0 | 56.7 | 49.2 | 41.2 | 34.5 | **29.6** | **23.5** | **18.1** |
| | **+ One-step adversary** | **0.737** | 61.8 | 55.9 | 49.5 | **43.3** | **37.2** | **31.7** | **25.7** | **19.8** |

**Overall training.** Combining the proposed loss with the standard Gaussian training (5) gives the full objective to minimize for our training method. For a given sample $(x, y) \sim P$, and by letting $L^{\mathtt{nat}} := \mathbb{E}_\delta [\mathcal{L}(F(x + \delta), y)]$, the final loss of SmoothMix is given by:

$$L := L^{\mathtt{nat}} + \eta \cdot L^{\mathtt{mix}} \qquad (12)$$

where $\eta > 0$ is a hyperparameter to control the trade-off between accuracy and robustness. Algorithm 1 in Appendix A demonstrates a concrete training procedure of SmoothMix using $m$ samples of $\delta$ for the Monte Carlo approximation.

## 4. Experiments

We evaluate the effectiveness of our method extensively on MNIST (LeCun et al., 1998), CIFAR-10 (Krizhevsky, 2009), and ImageNet (Russakovsky et al., 2015) datasets.[2] Overall, the results consistently highlight that our newly proposed training can significantly improve the certified robustness of smoothed classifiers compared to existing robust training methods. We point out the improvements are especially remarkable on the certified accuracy at larger perturbations, at which SmoothMix mainly focus on compared to prior arts. We also conduct an extensive ablation study on the proposed method to convey a detailed analysis in Appendix H, verifying that our method (a) is robust to the choice of hyperparameters, and (b) is an effective way to control the robustness of smoothed classifiers against accuracy. The full details on the experimental setups, *e.g.*, baselines, evaluation metrics, and hyperparameters, are specified in Appendix B.

**Results on CIFAR-10.** We report the *approximate certified accuracy* and *average certified radius* (ACR) (see Appendix B.2) of smoothed classifiers from ResNet-110

(He et al., 2016) using the full CIFAR-10 test dataset. We consider three different models as varying the noise level $\sigma \in \{0.25, 0.5, 1.0\}$ (the results with $\sigma = 1.0$ can be found in Appendix E). When SmoothMix is used, we consider fixed hyperparameters of $T = 4$, $m = 2$, and $\eta = 5.0$ throughout the experiments. We make sure that $\alpha \cdot T$ to be proportional to $\sigma$: there are different statistical upper bounds on the certified radius depending on $\sigma$. Namely, we set $\alpha = 0.5, 1.0, 2.0$ with $\sigma = 0.25, 0.5, 1.0$, respectively.

The results are summarized in Table 1 (and Figure 4 in Appendix). We observe that our method generally exhibits better trade-offs between accuracy and certified robustness compared to other baselines: *e.g.*, at $\sigma = 0.5$, "SmoothMix" could improve the previous best result from "Consistency" by a significant margin of $0.720 \rightarrow 0.737$. Without the single-step adversary, "SmoothMix" can effectively preserve the clean accuracy while also improving ACR, *e.g.*, at $\sigma = 0.25$, "SmoothMix" could even improve the clean accuracy of "Gaussian": although "MACER" could improve the clean accuray as well, one could see that their improvements in robust accuracy are relatively limited.

## 5. Conclusion

In this paper, we observe that adversarial training with an *unrestricted* adversary can be feasible and even more promising (compared to the *restricted* ones) when it comes with smoothed classifiers, based on a close relationship between randomized smoothing and *confidence-calibrated* classifiers (Guo et al., 2017; Lee et al., 2018). Although our focus in this paper is currently limited only to the over-confidence issue, we believe there are still many rooms to be explored in future for another such connection, *e.g.*, could the recent advances in the literature of uncertainty estimation of deep neural networks (Hendrycks et al., 2019; Tack et al., 2020) help to improve the robustness of smoothed classifiers.

---

[2] All the experimental results on MNIST and ImageNet datasets are reported in Appendix D and F, respectively.

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

# A. Training procedure of SmoothMix

---

**Algorithm 1** SmoothMix training

---

**Input:** Sample $(x, y) \sim P$. smoothing factor $\sigma$. number of noise samples $m$. number of steps $T$. step size $\alpha$. regularization strength $\eta > 0$.

---

1: Sample $\delta_1, \cdots, \delta_m \sim \mathcal{N}(0, \sigma^2 I)$, and $\lambda \sim \mathcal{U}\left(\left[0, \frac{1}{2}\right]\right)$
2: // FIND AN ADVERSARIAL EXAMPLE
3: $\tilde{x}^{(0)}, \hat{F}(x^{(0)}) \leftarrow x, \frac{1}{m} \sum_{i=1}^{m} F(x + \delta_i)$
4: **for** $t = 0$ **to** $T - 1$ **do**
5: $\quad J(\tilde{x}^{(t)}) \leftarrow -\log \hat{F}_y(\tilde{x}^{(t)})$
6: $\quad \tilde{x}^{(t+1)} \leftarrow \tilde{x}^{(t)} + \alpha \cdot \frac{\nabla_x J(\tilde{x}^{(t)})}{\|\nabla_x J(\tilde{x}^{(t)})\|_2}$
7: $\quad \hat{F}(\tilde{x}^{(t+1)}) \leftarrow \frac{1}{m} \sum_{i=1}^{m} F(\tilde{x}^{(t+1)} + \delta_i)$
8: **end for**
9: **if** use_single_step **then** $x \leftarrow \tilde{x}^{(1)}$
10: // COMPUTE THE SMOOTHMIX LOSS
11: $x^{\mathtt{mix}}, y^{\mathtt{mix}} \leftarrow ((1 - \lambda) \cdot x + \lambda \cdot \tilde{x}^{(T)}), ((1 - \lambda) \cdot \hat{F}(x) + \lambda \cdot \frac{1}{C})$
12: **for** $i = 1$ **to** $m$ **do**
13: $\quad L_i^{\mathtt{nat}}, L_i^{\mathtt{mix}} \leftarrow \mathcal{L}(F(x + \delta_i), y), \mathcal{L}(F(x^{\mathtt{mix}} + \delta_i), y^{\mathtt{mix}})$
14: **end for**
15: $L \leftarrow \frac{1}{m} \sum_i (L_i^{\mathtt{nat}} + \eta \cdot L_i^{\mathtt{mix}})$

---

# B. Experimental details

Throughout our experiments, we follow the same training details of prior works (Cohen et al., 2019; Salman et al., 2019; Zhai et al., 2020; Jeong & Shin, 2020) for a fair comparison: more specifically, we use LeNet (LeCun et al., 1998) for MNIST, ResNet-110 (He et al., 2016) for CIFAR-10, and ResNet-50 (He et al., 2016) for ImageNet. We train every model via stochastic gradient descent using Nesterov momentum of weight 0.9 without dampening. We set a weight decay of $10^{-4}$ for all the models. We consider three different noise levels $\sigma \in \{0.25, 0.5, 1.0\}$ for smoothing classifiers for MNIST and CIFAR-10 models, and $\sigma \in \{0.5, 1.0\}$ in the case of ImageNet. We used up to 4 NVIDIA TITAN Xp GPUs to run each configurations considered in our experiments, both for training and certification: more specifically, we used a single GPU to run every experimenet on MNIST and CIFAR-10, and four GPUs to run ImageNet models.

## B.1. Baseline methods

We compare our method with a variety of existing techniques proposed for a robust training of smoothed classifiers, as listed in what follows: (a) Gaussian (Cohen et al., 2019): standard training with Gaussian augmentation; (b) Stability training (Li et al., 2019): a cross-entropy regularization between $F(x)$ and $F(x + \delta)$; (c) SmoothAdv (Salman et al., 2019): adversarial training on smoothed classifier; (d) MACER (Zhai et al., 2020): a regularization that maximizes an approximative form of the certified radius (3); and (e) Consistency (Jeong & Shin, 2020): a KL-divergence based regularization that minimizes the variance of $F(x + \delta)$ across $\delta$. Whenever possible, we use the pre-trained models released by authors for our evaluation to reproduce the baselines: *e.g.*, for CIFAR-10 results of SmoothAdv, we report the performance evaluated from the pre-trained models released by the authors[3] for a fixed configuration of $T = 10, \varepsilon = 1.0$, and $m = 8$.

## B.2. Evaluation metrics

Our evaluation of the robustness for a given smoothed classifier $\hat{f}$ is largely based on the protocol proposed by Cohen et al. (2019), similarly to prior works (Salman et al., 2019; Zhai et al., 2020; Jeong & Shin, 2020): more concretely, Cohen et al. (2019) proposed a practical Monte Carlo based certification procedure, namely CERTIFY, that returns the prediction of $\hat{f}$ and a "safe" lower bound of certified radius over the randomness of $n$ samples with probability at least $1 - \alpha$, or abstains the certification.

---

[3] https://github.com/Hadisalman/smoothing-adversarial

From CERTIFY, we consider two evaluation metrics: (a) the *approximate certified test accuracy* at various radii: the fraction of the test dataset which CERTIFY classifies correctly with radius larger than $r$ without abstaining, and (b) the *average certified radius* (ACR) (Zhai et al., 2020): the average of certified radii returned by CERTIFY on the test dataset counting only the correctly classified samples, namely $\text{ACR} := \frac{1}{|\mathcal{D}_{\text{test}}|} \sum_{(x,y) \in \mathcal{D}_{\text{test}}} \text{CR}(f, \sigma, x) \cdot \mathbf{1}_{\hat{f}(x)=y}$, where $\mathcal{D}_{\text{test}}$ is the test dataset, and CR denotes the certified radius from $\text{CERTIFY}(f, \sigma, x)$. Here, the latter metric, ACR, is for a better comparison of robustness under trade-off between accuracy and robustness (Tsipras et al., 2019; Zhang et al., 2019). We use the official PyTorch implementation[4] of CERTIFY, with $n = 100,000$, $n_0 = 100$ and $\alpha = 0.001$, following (Cohen et al., 2019; Salman et al., 2019; Jeong & Shin, 2020).

### B.3. Datasets

**MNIST** dataset (LeCun et al., 1998) consists 70,000 gray-scale hand-written digit images of size 28×28, 60,000 for training and 10,000 for testing. Each of the images is labeled from 0 to 9, i.e., there are 10 classes. We do not perform any pre-processing except for normalizing the range of each pixel from 0-255 to 0-1. When MNIST is used for training, we use LeNet (LeCun et al., 1998) for 90 epochs and use the initial learning rate of 0.01. The learning rate is decayed by 0.1 at 30-th and 60-th epoch.

**CIFAR-10** dataset (Krizhevsky, 2009) consist of 60,000 RGB images of size 32×32 pixels, 50,000 for training and 10,000 for testing. Each of the images is labeled to one of 10 classes, and the number of data per class is set evenly, i.e., 6,000 images per each class. We use the standard data-augmentation scheme of random horizontal flip and random translation up to 4 pixels, as also used by other baselines (Cohen et al., 2019; Salman et al., 2019; Zhai et al., 2020; Jeong & Shin, 2020). We also normalize the images in pixel-wise by the mean and the standard deviation calculated from the training set. When CIFAR-10 is used for training, we train ResNet-110 (He et al., 2016) models for 150 epochs with initial learning rate of 0.1. The learning rate if decated by 0.1 at 50-th and 100-th epoch.

**ImageNet** classification dataset (Russakovsky et al., 2015) consists of 1.2 million training images and 50,000 validation images, which are labeled by one of 1,000 classes. For data-augmentation, we perform 224×224 random cropping with random resizing and horizontal flipping to the training images. At test time, on the other hand, 224×224 center cropping is performed after re-scaling the images into 256×256. When ImageNet is used for training, we train ResNet-50 (He et al., 2016) models for 90 epochs with initial learning rate of 0.1. The learning rate if decated by 0.1 at 30-th and 60-th epoch.

### B.4. Detailed hyperparameters for baselines

**Stability training (Li et al., 2019)** uses a single hyperparameter $\lambda > 0$ to control the relative strength of the stability regularization compared to the standard cross-entropy loss. In our experiments, we use $\lambda = 2$ by default for this method, but except for the "$\sigma = 1.0$" model on CIFAR-10: in this case, we had to reduce it to $\lambda = 1$ for a stable training.

**SmoothAdv (Salman et al., 2019)** mainly controls three hyperparameters those are for performing projected gradient descent (PGD) to find adversarial examples in the training: namely, it uses $m$: the number of noise samples, $T$: the number of PGD steps, and $\varepsilon$: an $\ell_2$-norm restriction on adversarial perturbations. For SmoothAdv models, we fix $T = 10$ and $\varepsilon = 1.0$ throughout the experiments. In case of $m$, and use $m = 4$ for MNIST models, and $m = 8$ for CIFAR-10. Following Salman et al. (2019), we also adopt the *warm-up* strategy on $\varepsilon$, *i.e.*, it is initially set to zero, and gradually increased for the first 10 epochs up to the original value of $\varepsilon$.

**MACER (Zhai et al., 2020)** adds four hyperparameters to the training: namely, it uses $m$: the number of noise samples, $\lambda$: the relative strength of regularization, $\beta$: a temperature scaling factor, and $\gamma$: a margin gap. We follow the configurations reported by Zhai et al. (2020) to reproduce the MNIST results: namely, we use $m = 16$, $\beta = 16.0$, $\gamma = 8.0$ and $\lambda = 16.0$. We use $\lambda = 6.0$ in case of $\sigma = 1.0$ on MNIST, however, for a better training stability. We use the pre-trained models released by the authors for evaluations on CIFAR-10, which can be downloaded at `https://github.com/RuntianZ/macer`. These CIFAR-10 models are reported to be trained with $m = 16$, $\beta = 16.0$, $\gamma = 8.0$, and $\lambda = 12.0$ and 4.0 for $\sigma = 0.25$ and 0.5, respectively. For $\sigma = 1.0$, $\lambda$ is initially set to 0, and changed to $\lambda = 12.0$ after the first learning rate decay.

**Consistency (Jeong & Shin, 2020)** controls two hyperparameters, namely $\lambda$ and $\eta$, each for the relative strength of the consistency term and the entropy term, respectively. We obtain results from the best hyperparameters those reported by Jeong & Shin (2020) when the consistency regularization is applied to the Gaussian training baseline, both in MNIST and

---

[4]`https://github.com/locuslab/smoothing`

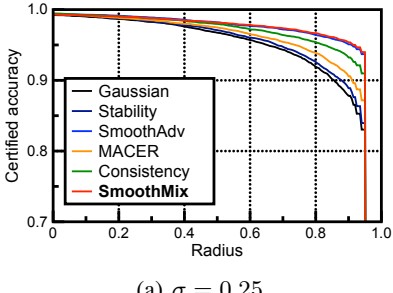 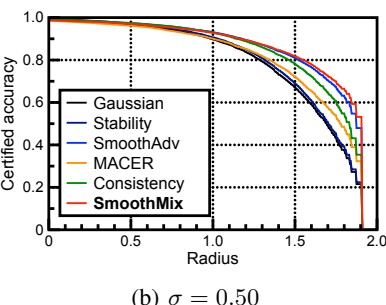 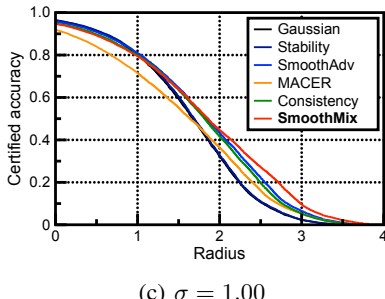

|  (a) $\sigma = 0.25$ | (b) $\sigma = 0.50$ | (c) $\sigma = 1.00$ |

*Figure 3.* Comparison of approximate certified accuracy for various training methods on MNIST. The sharp drop of certified accuracy in each plots is due to that there is a strict upper bound in radius that CERTIFY can output for a given $\sigma$ and $n = 100,000$.

CIFAR-10 datasets. More concretely, we fix $\eta = 0.5$ for every model, and use $\lambda = 5$ for MNIST and $\lambda = 10$ for CIFAR-10 models by default. In case of $\sigma = 0.25$, $\lambda$ is doubled in both datasets, *i.e.*, $\lambda = 10$ and $\lambda = 20$ for MNIST and CIFAR-10, respectively, as it is shown to achieve better ACRs.

## C. Related work

**Certified adversarial robustness.** We focus on improving adversarial robustness of *randomized smoothing* (Cohen et al., 2019) based classifiers, which is currently one of prominent ways to obtain a classifier with a robustness certification. In general, there have been many attempts other than randomized smoothing to provide a robustness certification of deep neural networks (Gehr et al., 2018; Wong & Kolter, 2018; Mirman et al., 2018; Xiao et al., 2019; Gowal et al., 2019; Zhang et al., 2020a), and correspondingly with attempts to further improve the robustness with respect to those certification protocols (Croce et al., 2019; Croce & Hein, 2020; Balunovic & Vechev, 2020). Nevertheless, randomized smoothing has attracted particular attention as the first approach that could successfully scaled up to the ImageNet dataset (Russakovsky et al., 2015). A more complete taxonomy on the literature can be found in Li et al. (2020).

**Confidence-calibrated training.** *Overconfident predictions* of deep neural networks (Pereyra et al., 2017) have been considered as problematic in many scenarios, *e.g.*, uncertainty estimation of in-distribution samples (Guo et al., 2017; Jiang et al., 2018; Kumar et al., 2019), those of out-of-distribution samples (Hendrycks & Gimpel, 2017; Lee et al., 2018; Meinke & Hein, 2020), and ensemble learning (Lee et al., 2017), just to name a few. In the context of adversarial training, Stutz et al. (2020) have shown that regularizing confidence on adversarial examples to be uniform can improve detection of adversarial examples from unseen threat models. In this paper, we address the overconfidence at adversarial examples particularly focusing on *smoothed classifiers*, observing that a simple approach of directly fixing the problem could significantly improve the certified robustness.

**Mixup-based training.** Originally, *mixup* (Zhang et al., 2018) has proposed as a simple yet effective data augmentation scheme to improve generalization and robustness (against small adversarial attacks) of deep neural networks, and there have been significant follow-up works to further improve this form (Verma et al., 2019; Yun et al., 2019; Kim et al., 2020; 2021). Recently, Zhang et al. (2021) have also explored on theoretical justifications behind how could such an augmentation improves generalization and robustness. Although our method uses a similar linear interpolation scheme of mixup, there is still an essential difference between ours and this line of works: namely, we do not rely on the prior of interpolating two (or more) *independent* samples, but rather aims to directly calibrate predictions between a clean and its (unrestricted) adversarial example, *i.e.*, we consider a new form of *self-mixup* training.

There have been also attempts to employ mixup particularly for improving adversarial robustness: Lamb et al. (2019) have shown that an additional mixup loss between adversarial examples upon the standard mixup training achieves a comparable robustness to adversarial training (AT) (Madry et al., 2018), while not compromising the clean accuracy as much as AT; Lee et al. (2020) have proposed *Adversarial Vertex Mixup* to improve AT, by extrapolating predictions along the direction of adversarial perturbation up to few times of its norm via mixup training. Our proposed method can be differentiated to these approaches, in a sense that we employ mixup not to directly improve the robustness of a given neural network, but of its smoothed counterpart. It is also our unique perspective that we consider *unrestricted* adversarial examples to be interpolated.

*Table 2.* Comparison of approximate certified test accuracy (%) and ACR on MNIST. All the models are trained and evaluated with the same smoothing factor specified by $\sigma$. Each value except ACR indicates the fraction of test samples those have $\ell_2$ certified radius larger than the threshold specified at the top row. We set our results bold-faced whenever the value improves the Gaussian baseline, and underlined whenever the value improves the best among the considered baselines.

| $\sigma$ | Models (MNIST) | ACR | 0.00 | 0.25 | 0.50 | 0.75 | 1.00 | 1.25 | 1.50 | 1.75 | 2.00 | 2.25 | 2.50 | 2.75 |
|---|---|---|---|---|---|---|---|---|---|---|---|---|---|---|
| | Gaussian (Cohen et al., 2019) | 0.911 | 99.2 | 98.5 | 96.7 | 93.3 | 0.0 | 0.0 | 0.0 | 0.0 | 0.0 | 0.0 | 0.0 | 0.0 |
| | Stability training (Li et al., 2019) | 0.915 | 99.3 | 98.6 | 97.1 | 93.8 | 0.0 | 0.0 | 0.0 | 0.0 | 0.0 | 0.0 | 0.0 | 0.0 |
| | SmoothAdv (Salman et al., 2019) | 0.932 | 99.4 | 99.0 | 98.2 | 96.8 | 0.0 | 0.0 | 0.0 | 0.0 | 0.0 | 0.0 | 0.0 | 0.0 |
| | MACER (Zhai et al., 2020) | 0.920 | 99.3 | 98.7 | 97.5 | 94.8 | 0.0 | 0.0 | 0.0 | 0.0 | 0.0 | 0.0 | 0.0 | 0.0 |
| 0.25 | Consistency (Jeong & Shin, 2020) | 0.928 | 99.5 | 98.9 | 98.0 | 96.0 | 0.0 | 0.0 | 0.0 | 0.0 | 0.0 | 0.0 | 0.0 | 0.0 |
| | **SmoothMix** ($\eta = 1.0$) | **0.931** | **99.5** | 98.9 | **98.2** | 96.4 | 0.0 | 0.0 | 0.0 | 0.0 | 0.0 | 0.0 | 0.0 | 0.0 |
| | **+ One-step adversary** | **0.933** | 99.4 | **99.0** | **98.2** | **96.9** | 0.0 | 0.0 | 0.0 | 0.0 | 0.0 | 0.0 | 0.0 | 0.0 |
| | **SmoothMix** ($\eta = 5.0$) | **0.932** | 99.4 | **99.0** | **98.2** | 96.7 | 0.0 | 0.0 | 0.0 | 0.0 | 0.0 | 0.0 | 0.0 | 0.0 |
| | **+ One-step adversary** | **0.933** | 99.3 | **99.0** | **98.2** | **97.0** | 0.0 | 0.0 | 0.0 | 0.0 | 0.0 | 0.0 | 0.0 | 0.0 |
| | Gaussian (Cohen et al., 2019) | 1.553 | 99.2 | 98.3 | 96.8 | 94.3 | 89.7 | 81.9 | 67.3 | 43.6 | 0.0 | 0.0 | 0.0 | 0.0 |
| | Stability training (Li et al., 2019) | 1.570 | 99.2 | 98.5 | 97.1 | 94.8 | 90.7 | 83.2 | 69.2 | 45.4 | 0.0 | 0.0 | 0.0 | 0.0 |
| | SmoothAdv (Salman et al., 2019) | 1.687 | 99.0 | 98.3 | 97.3 | 95.8 | 93.2 | 88.5 | 81.1 | 67.5 | 0.0 | 0.0 | 0.0 | 0.0 |
| | MACER (Zhai et al., 2020) | 1.594 | 98.5 | 97.5 | 96.2 | 93.7 | 90.0 | 83.7 | 72.2 | 54.0 | 0.0 | 0.0 | 0.0 | 0.0 |
| 0.50 | Consistency (Jeong & Shin, 2020) | 1.657 | 99.2 | 98.6 | 97.6 | 95.9 | 93.0 | 87.8 | 78.5 | 60.5 | 0.0 | 0.0 | 0.0 | 0.0 |
| | **SmoothMix** ($\eta = 1.0$) | **1.678** | 99.0 | **98.4** | **97.4** | **95.7** | **93.0** | **88.1** | **80.0** | **65.6** | 0.0 | 0.0 | 0.0 | 0.0 |
| | **+ One-step adversary** | **1.694** | 98.8 | 98.1 | **97.1** | 95.3 | 92.7 | **88.3** | **81.7** | **69.5** | 0.0 | 0.0 | 0.0 | 0.0 |
| | **SmoothMix** ($\eta = 5.0$) | **1.694** | 98.7 | 98.0 | **97.0** | 95.3 | 92.7 | **88.5** | **81.8** | **70.0** | 0.0 | 0.0 | 0.0 | 0.0 |
| | **+ One-step adversary** | **1.685** | 98.2 | 97.5 | 96.3 | 94.5 | 91.3 | 87.4 | 81.0 | **70.7** | 0.0 | 0.0 | 0.0 | 0.0 |
| | Gaussian (Cohen et al., 2019) | 1.620 | 96.3 | 94.4 | 91.4 | 86.8 | 79.8 | 70.9 | 59.4 | 46.2 | 32.5 | 19.7 | 10.9 | 5.8 |
| | Stability training (Li et al., 2019) | 1.634 | 96.5 | 94.6 | 91.6 | 87.2 | 80.7 | 71.7 | 60.5 | 47.0 | 33.4 | 20.6 | 11.2 | 5.9 |
| | SmoothAdv (Salman et al., 2019) | 1.779 | 95.8 | 93.9 | 90.6 | 86.5 | 80.8 | 73.7 | 64.6 | 53.9 | 43.3 | 32.8 | 22.2 | 12.1 |
| | MACER (Zhai et al., 2020) | 1.598 | 91.6 | 88.1 | 83.5 | 77.7 | 71.1 | 63.7 | 55.7 | 46.8 | 38.4 | 29.2 | 20.0 | 11.5 |
| 1.00 | Consistency (Jeong & Shin, 2020) | 1.740 | 95.0 | 93.0 | 89.7 | 85.4 | 79.7 | 72.7 | 63.6 | 53.0 | 41.7 | 30.8 | 20.3 | 10.7 |
| | **SmoothMix** ($\eta = 1.0$) | **1.788** | 95.5 | 93.5 | 90.5 | 86.2 | **80.6** | **73.4** | **64.3** | 53.7 | 43.2 | **33.5** | **23.9** | **14.1** |
| | **+ One-step adversary** | **1.816** | 94.7 | 92.4 | 89.2 | 84.6 | 79.4 | **72.5** | **64.0** | **54.5** | **44.8** | **36.2** | **27.4** | **18.7** |
| | **SmoothMix** ($\eta = 5.0$) | **1.820** | 93.7 | 91.6 | 88.1 | 83.5 | 77.9 | 70.9 | **62.7** | 53.8 | **44.8** | **36.6** | **28.9** | **21.5** |
| | **+ One-step adversary** | **1.823** | 93.3 | 90.9 | 87.5 | 83.0 | 77.5 | 70.6 | **62.7** | 53.4 | **44.9** | **37.1** | **29.3** | **22.4** |

# D. Results on MNIST

For MNIST (LeCun et al., 1998) experiments, we report the approximate certified accuracy and ACR of smoothed classifiers obtained from LeNet (LeCun et al., 1998) with different training methods, including SmoothMix, using the full MNIST test dataset. We consider three different models as varying the noise level $\sigma \in \{0.25, 0.5, 1.0\}$. During inference, we apply randomized smoothing with the same $\sigma$ used in the training. When SmoothMix is used, we consider a fixed hyperparameter value for $\alpha = 1.0$ and $m = 4$, the step size and the number of noise samples. We set $T = 2, 4, 8$ for the models with $\sigma = 0.25, 0.5, 1.0$, respectively, based on our empirical observation that it is beneficial to set $\alpha \cdot T$ to be proportional to $\sigma$. We apply the same $m = 4$ for SmoothAdv, *i.e.*, for adversarial training, and $T = 10$ with an $\ell_2$-ball of radius $\varepsilon = 1.0$.

The results are presented in Table 2 and Figure 3. Overall, we observe that our proposed SmoothMix loss (11) added to the Gaussian training dramatically improve the certified test accuracy from "Gaussian". By considering the one-step adversary (Section 3.2) in training, we could further improve the robust accuracy, significantly improving ACRs compared to the previous state-of-the-art training methods: *e.g.*, our method could improve ACRs with $\sigma = 1.0$ from $1.779 \rightarrow 1.823$. This shows that improvements from SmoothMix can be orthogonal to those from SmoothAdv. It is also remarkable that even without the one-step adversarial example, one could further improve the certified robustness by simply increasing the relative strength $\eta$ of the SmoothMix loss, *e.g.*, by $1.0 \rightarrow 5.0$ as presented in Table 2: *e.g.*, "SmoothMix" with $\eta = 5.0$ still outperforms "SmoothAdv" by $1.779 \rightarrow 1.820$ at $\sigma = 1.0$. Finally, we note that our models could substantially improve the robustness at larger perturbations with less degradation in the clean accuracy, *e.g.*, compared to "MACER" or "Consistency": considering that they are also regularization based approaches to control the robustness via controlling their regularization strength, the results show that our form of loss could better compensate the trade-off between accuracy and robustness.

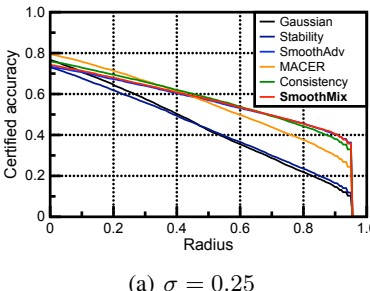

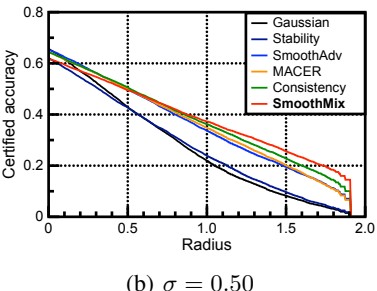

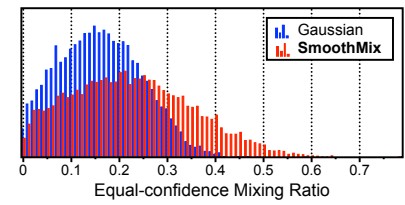

(a) $\sigma = 0.25$         (b) $\sigma = 0.50$

*Figure 4.* Comparison of approximate certified accuracy for various training methods on CIFAR-10. The sharp drop of certified accuracy in each plots is due to that there is a strict upper bound in radius that CERTIFY can output for a given $\sigma$ and $n = 100,000$.

*Figure 5. Equal-confidence mixing ratios* on CIFAR-10, *i.e.*, the minimal mixing ratios for changing the correct prediction when each input is linearly interpolated to its adversarial example.

## E. Additional results on CIFAR-10

In this section, we report additional experimental results on CIFAR-10 (Krizhevsky, 2009), namely with $\sigma = 1.0$ (see Table 1 for the results for $\sigma \in \{0.25, 0.5\}$). We defer this results to Appendix as the scenario can be less practical compared to the others: *e.g.*, the clean accuracy in this setup is $< 50\%$ in most cases. We follow the same experimental details as specified in Section 4 and Appendix B, including the common hyperparameter choice of $\eta = 5.0$ for SmoothMix for other experiments as well. Again, we compare our method with various existing robust training methods for smoothed classifiers (Cohen et al., 2019; Li et al., 2019; Salman et al., 2019; Zhai et al., 2020; Jeong & Shin, 2020), and Table 3 summarizes the results. Overall, we still observe a similar trend to Section 4 that (a) "SmoothMix" offers a significant improvement of robust accuracy without compromising the clean accuracy much, and (b) incorporating the one-step adversary thus can further complementarily boost ACR to outperform other state-of-the-art baseline training methods: *e.g.*, it is notable that "SmoothMix + One-step adversary" achieves fairly comparable or better robust accuracy than MACER while maintaining much higher clean accuracy, *i.e.*, the certified test accuracy at $r = 0.0$, namely $41.4 \rightarrow 45.1$. This confirms that our proposed SmoothMix can offer a better trade-off between accuracy and certified robustness during training.

*Table 3.* Comparison of approximate certified test accuracy (%) and ACR on CIFAR-10. All the models are trained and evaluated with the same smoothing factor specified by $\sigma$. Each value except ACR indicates the fraction of test samples those have $\ell_2$ certified radius larger than the threshold specified at the top row. We set our results bold-faced whenever the value improves the Gaussian baseline, and underlined whenever the value achieves the best among the considered baselines. * indicates that the results are evaluated from the official pre-trained models released by authors.

| $\sigma$ | Models (CIFAR-10) | ACR | 0.00 | 0.25 | 0.50 | 0.75 | 1.00 | 1.25 | 1.50 | 1.75 | 2.00 | 2.25 |
|---|---|---|---|---|---|---|---|---|---|---|---|---|
| | Gaussian (Cohen et al., 2019) | 0.542 | 47.2 | 39.2 | 34.0 | 27.8 | 21.6 | 17.4 | 14.0 | 11.8 | 10.0 | 7.6 |
| | Stability training (Li et al., 2019) | 0.526 | 43.5 | 38.9 | 32.8 | 27.0 | 23.1 | 19.1 | 15.4 | 11.3 | 7.8 | 5.7 |
| | SmoothAdv* (Salman et al., 2019) | 0.660 | 50.8 | 44.9 | 39.0 | 33.6 | 28.5 | 23.7 | 19.4 | 15.4 | 12.0 | 8.7 |
| 1.00 | MACER* (Zhai et al., 2020) | 0.744 | 41.4 | 38.5 | 35.2 | 32.3 | 29.3 | 26.4 | 23.4 | 20.2 | 17.4 | 14.5 |
| | Consistency (Jeong & Shin, 2020) | 0.756 | 46.3 | 42.2 | 38.1 | 34.3 | 30.0 | 26.3 | 22.9 | 19.7 | 16.6 | 13.8 |
| | **SmoothMix (Ours)** | **0.725** | 47.1 | **42.5** | **37.5** | **32.9** | **28.7** | **24.9** | **21.3** | **18.3** | **15.5** | **12.6** |
| | **+ One-step adversary** | **0.773** | 45.1 | **41.5** | **37.5** | **33.8** | **30.2** | **26.7** | **23.4** | **20.2** | 17.2 | **14.7** |

## F. Results on ImageNet

We also compare our method on ImageNet (Russakovsky et al., 2015) classification dataset, to verify the scalability of the method on large-scale datasets. In this experiment, we perform our evaluation on the sub-sampled validation dataset of ImageNet with 500 samples following the previous works (Cohen et al., 2019; Salman et al., 2019; Jeong & Shin, 2020). When SmoothMix is used, we simply set $T = 1$ and $m = 1$ mainly in order to reduce the overall training cost, and we fix $\alpha = 8.0$ for both cases of $\sigma = 0.5, 1.0$: this choice leads larger $\alpha \cdot T$ when $\sigma = 0.5$ compared to the MNIST and CIFAR-10 experiments, but we empirically observe that ImageNet is less sensitive to $\alpha \cdot T$, possibly due to that ImageNet consists of higher-resolution inputs, *i.e.*, higher input dimension accordingly, than the others. We use the one-step adversary (Section 3.2) by default here, but we make sure that each adversarial example (found with a large $\alpha$) is further projected in a

$\ell_2$-ball of $\epsilon = 1.0$ before it replaces the clean sample, which can be done without adding significant computational overhead. Table 4 summarizes the results, and we still observe the effectiveness of SmoothMix compared to the baseline methods, both in terms of ACR and certified test accuracy.

*Table 4.* Comparison of approximate certified test accuracy (%) on ImageNet. We set our results bold-faced whenever the value improves the Gaussian baseline, and underlined whenever the value achieves the best among the considered baselines.

| $\sigma$ | Models (ImageNet) | ACR | 0.0 | 0.5 | 1.0 | 1.5 | 2.0 | 2.5 | 3.0 | 3.5 |
|---|---|---|---|---|---|---|---|---|---|---|
| 0.50 | Gaussian (Cohen et al., 2019) | 0.733 | 57 | 46 | 37 | 29 | 0 | 0 | 0 | 0 |
| | Consistency (Jeong & Shin, 2020) | 0.822 | 55 | 50 | 44 | 34 | 0 | 0 | 0 | 0 |
| | SmoothAdv (Salman et al., 2019) | 0.825 | 54 | 49 | 43 | 37 | 0 | 0 | 0 | 0 |
| | **SmoothMix (Ours)** | **0.846** | 55 | **50** | **43** | **38** | 0 | 0 | 0 | 0 |
| 1.00 | Gaussian (Cohen et al., 2019) | 0.875 | 44 | 38 | 33 | 26 | 19 | 15 | 12 | 9 |
| | Consistency (Jeong & Shin, 2020) | 0.982 | 41 | 37 | 32 | 28 | 24 | 21 | 17 | 14 |
| | SmoothAdv (Salman et al., 2019) | 1.040 | 40 | 37 | 34 | 30 | 27 | 25 | 20 | 15 |
| | **SmoothMix (Ours)** | **1.047** | 40 | 37 | **34** | **30** | **26** | **24** | **20** | **17** |

# G. Variance of results over multiple runs

In our experiments, we report single-run results for ACR and certified robust accuracy as also done by (Cohen et al., 2019; Salman et al., 2019; Li et al., 2019; Zhai et al., 2020; Jeong & Shin, 2020), considering that ACR is fairly a robust metric to network initialization: *e.g.*, in Table 5, we report the mean and standard deviation of ACRs across 5 seeds for the MNIST results reported in Table 2. Overall, we confirm that ACR generally shows low variance over multiple runs across a wide range of training methods, including ours.

*Table 5.* Comparison of ACR for various training methods on MNIST. The reported values are the mean and standard deviation across 5 seeds. We set our result bold-faced whenever the value improves the baseline, and the underlined are best-performing model per $\sigma$.

| ACR (MNIST) | $\sigma = 0.25$ | $\sigma = 0.50$ | $\sigma = 1.00$ |
|---|---|---|---|
| Gaussian (Cohen et al., 2019) | $0.9108_{\pm 0.0003}$ | $1.5581_{\pm 0.0016}$ | $1.6184_{\pm 0.0021}$ |
| Stability (Li et al., 2019) | $0.9152_{\pm 0.0007}$ | $1.5719_{\pm 0.0028}$ | $1.6341_{\pm 0.0018}$ |
| SmoothAdv (Salman et al., 2019) | $0.9322_{\pm 0.0005}$ | $1.6872_{\pm 0.0007}$ | $1.7786_{\pm 0.0017}$ |
| MACER (Zhai et al., 2020) | $0.9201_{\pm 0.0006}$ | $1.5899_{\pm 0.0069}$ | $1.5950_{\pm 0.0051}$ |
| Consistency (Jeong & Shin, 2020) | $0.9279_{\pm 0.0003}$ | $1.6549_{\pm 0.0011}$ | $1.7376_{\pm 0.0017}$ |
| **SmoothMix ($\eta = 1.0$)** | **$0.9296_{\pm 0.0003}$** | **$1.6776_{\pm 0.0007}$** | **$1.7867_{\pm 0.0020}$** |
| **+ One-Step adversary** | **$0.9330_{\pm 0.0004}$** | **$1.6932_{\pm 0.0009}$** | **$1.8169_{\pm 0.0011}$** |
| **SmoothMix ($\eta = 5.0$)** | **$0.9317_{\pm 0.0002}$** | **$1.6932_{\pm 0.0007}$** | **$1.8185_{\pm 0.0016}$** |
| **+ One-Step adversary** | **$0.9332_{\pm 0.0002}$** | **$1.6851_{\pm 0.0003}$** | **$1.8212_{\pm 0.0013}$** |

# H. Ablation study

We also conduct an ablation study to investigate the individual effects of the hyperparameters in our method. Unless otherwise noted, we perform experiments on MNIST with $\sigma = 1.0$. We report the detailed values of the results for Figure 6(a), 6(b), 6(c) and 6(d) in Table 6, 7, 8 and 9, respectively.

**Effect of $\eta$.** By design, SmoothMix controls the trade-off between accuracy and robustness by adjusting $\eta$, the relative strength of $L^{\texttt{mix}}$ over $L^{\texttt{nat}}$ (12). Here, we further examine the effect of $\eta$ by comparing the certified robustness on varying $\eta \in \{1, 2, 4, 8, 16\}$: the results in Figure 6(a) show that increasing $\eta$ consistently improves the certified robustness of the classifier, which confirms $L^{\texttt{mix}}$, the mixup loss, as an effective term to trade-off the robustness against $L^{\texttt{nat}}$ for accuracy.

**Trade-off between $\alpha$ and $T$.** In practice, SmoothMix can trade-off between the step size $\alpha$ and the number of steps $T$ to compensate between a more accurate optimization of (6) and its computational cost, while maintaining the effective range of the perturbation by $\alpha \cdot T$. Figure 6(b) explores this trade-off, by comparing models trained with different combinations of $(\alpha, T)$ under control of $\alpha \cdot T = 8.0$. Interestingly, the results indicate that the choice of $\alpha$ and $T$ does not significantly affect the final performance as long as $\alpha \cdot T$ is constant: all the considered combinations achieve similar robustness, with only a

(a) Effect of $\eta$      (b) Effect of $\alpha$ and $T$      (c) Restricted attack of $\varepsilon$      (d) Effect of $m$

*Figure 6.* Comparison of approximate certified test accuracy of SmoothMix and its ablations. "Gaussian" indicates the baseline training with Gaussian augmentation.

slight degradation in ACR even at $(\alpha, T) = (8.0, 1)$ (see Table 7). This suggests that (a) finding adversarial examples in a smoothed classifier can be simpler than one might expect, and (b) one can effectively reduce the training cost of SmoothMix using small $T$ in practice.

**Hard restriction on adversarial attacks.** One of key features of SmoothMix is at its *unrestricted* search of adversarial examples. Here, we examine the case when there is a hard restriction on each search, namely in $\ell_2$-radius of $\varepsilon \in \{2, 4, 6, 8\}$. The results presented in Figure 6(c) along with the Gaussian baseline ("Gaussian") and the original unrestricted setup ("$\varepsilon = \infty$") show that SmoothMix indeed works best when there is no such restrictions, although these ablations still reasonably improve the Gaussian baseline, *i.e.*, calibrating with adversarial examples outside the $\varepsilon$-ball can indeed help to improve the certified robustness in our training scheme.

**Effect of $m$.** Figure 6(d) investigates the effect of using different $m \in \{1, 2, 4, 8\}$, the number of noise samples to approximate the prediction of smoothed classifier: the larger $m$, the better approximation of smoothed classifier, which would be beneficial for both natural loss and SmoothMix loss (12). Overall, we observe that SmoothMix can still improve ACR from "Gaussian" even with $m = 1$, but with a moderate degradation in the clean accuracy: as $m$ is one of the crucial factors related to the total training cost in practice, one is recommended to use smaller $m$, *e.g.*, $m = 2$ or $4$, considering its little effect to the final ACR.

**Equal-confidence mixing ratios.** Recall from Figure 2 that we are motivated by the problem of *miscalibration* in smoothed classifiers between clean and its adversarial example. To see how much the proposed SmoothMix could alleviate this issue, we compare the distributions of the minimal mixing ratios that changes its prediction of a given classifier on the test dataset, namely the *equal-confidence mixing ratios*, before and after training with SmoothMix. For this comparison, we use ResNet-110 on CIFAR-10 assuming $\sigma = 0.25$. Figure 5 shows the result, and it indeed confirms SmoothMix has an effect of improving calibration between clean and adversarial examples.

*Table 6.* Comparison of ACR and approximate certified test accuracy on MNIST for varying $\eta$. We assume $\sigma = 1.0$ in this experiment. "Gaussian" indicates the baseline training with Gaussian augmentation. We set the results bold-faced whenever the value improves "Gaussian".

| Setups | ACR | 0.00 | 0.25 | 0.50 | 0.75 | 1.00 | 1.25 | 1.50 | 1.75 | 2.00 | 2.25 | 2.50 |
|---|---|---|---|---|---|---|---|---|---|---|---|---|
| Gaussian | 1.620 | 96.4 | 94.4 | 91.4 | 87.0 | 79.9 | 71.0 | 59.6 | 46.2 | 32.6 | 19.7 | 10.8 |
| $\eta = 1$ | **1.789** | 95.5 | 93.6 | 90.5 | 86.2 | **80.7** | **73.7** | **64.1** | **53.9** | **43.1** | **33.5** | **24.1** |
| $\eta = 2$ | **1.810** | 94.9 | 92.7 | 89.7 | 85.1 | 79.6 | **72.6** | **63.8** | **54.0** | **44.4** | **35.4** | **26.6** |
| $\eta = 4$ | **1.820** | 94.0 | 91.8 | 88.4 | 83.9 | 78.3 | **71.4** | **63.0** | **53.6** | **44.9** | **36.8** | **28.7** |
| $\eta = 8$ | **1.817** | 93.4 | 91.0 | 87.5 | 82.7 | 77.3 | 70.2 | **62.4** | **53.0** | **44.8** | **37.0** | **29.3** |
| $\eta = 16$ | **1.812** | 92.9 | 90.3 | 86.7 | 82.1 | 76.6 | 69.7 | **61.8** | **52.6** | **44.5** | **36.9** | **29.6** |

*Table 7.* Comparison of ACR and approximate certified test accuracy on MNIST for varying $\alpha$ and $T$ under control of $\alpha \cdot T = 8$. We assume $\sigma = 1.0$ in this experiment. "Gaussian" indicates the baseline training with Gaussian augmentation. We set the results bold-faced whenever the value improves "Gaussian".

| Setups | ACR | 0.00 | 0.25 | 0.50 | 0.75 | 1.00 | 1.25 | 1.50 | 1.75 | 2.00 | 2.25 | 2.50 |
|---|---|---|---|---|---|---|---|---|---|---|---|---|
| Gaussian | 1.620 | 96.4 | 94.4 | 91.4 | 87.0 | 79.9 | 71.0 | 59.6 | 46.2 | 32.6 | 19.7 | 10.8 |
| $(\alpha, T) = (8.0, 1)$ | **1.785** | 95.5 | 93.5 | 90.5 | 86.0 | **80.5** | **73.1** | **63.9** | **53.5** | **43.3** | **33.2** | **24.0** |
| $(\alpha, T) = (4.0, 2)$ | **1.788** | 95.4 | 93.4 | 90.4 | 85.9 | **80.5** | **73.5** | **63.9** | **53.5** | **43.1** | **33.4** | **24.4** |
| $(\alpha, T) = (2.0, 4)$ | **1.790** | 95.5 | 93.5 | 90.7 | 86.2 | **80.7** | **73.7** | **64.3** | **53.9** | **43.2** | **33.4** | **23.8** |
| $(\alpha, T) = (1.0, 8)$ | **1.789** | 95.5 | 93.6 | 90.5 | 86.2 | **80.7** | **73.7** | **64.1** | **53.9** | **43.1** | **33.5** | **24.1** |

*Table 8.* Comparison of ACR and approximate certified test accuracy on MNIST for varying $\varepsilon$, the hard limit on $\ell_2$-norm of adversarial perturbations. We assume $\sigma = 1.0$ in this experiment. "Gaussian" indicates the baseline training with Gaussian augmentation. "$\varepsilon = \infty$" denotes our original setup of unrestricted adversarial attacks. We set the results bold-faced whenever the value improves "Gaussian".

| Setups | ACR | 0.00 | 0.25 | 0.50 | 0.75 | 1.00 | 1.25 | 1.50 | 1.75 | 2.00 | 2.25 | 2.50 |
|---|---|---|---|---|---|---|---|---|---|---|---|---|
| Gaussian | 1.620 | 96.4 | 94.4 | 91.4 | 87.0 | 79.9 | 71.0 | 59.6 | 46.2 | 32.6 | 19.7 | 10.8 |
| $\varepsilon = 2.0$ | **1.723** | 96.1 | 94.3 | 91.4 | **87.1** | **81.2** | **73.6** | **63.7** | **52.1** | **39.8** | **28.2** | **16.6** |
| $\varepsilon = 4.0$ | **1.751** | 95.9 | 94.0 | 91.1 | 86.8 | **81.0** | **73.7** | **64.3** | **53.1** | **41.4** | **30.6** | **19.8** |
| $\varepsilon = 6.0$ | **1.778** | 95.6 | 93.7 | 90.6 | 86.5 | **80.8** | **73.7** | **64.4** | **53.8** | **42.8** | **32.6** | **22.9** |
| $\varepsilon = 8.0$ | **1.788** | 95.5 | 93.5 | 90.4 | 86.1 | **80.5** | **73.5** | **64.2** | **53.8** | **43.2** | **33.5** | **24.1** |
| $\varepsilon = \infty$ (Ours) | **1.789** | 95.5 | 93.6 | 90.5 | 86.2 | **80.7** | **73.7** | **64.1** | **53.9** | **43.1** | **33.5** | **24.1** |

*Table 9.* Comparison of ACR and approximate certified test accuracy on MNIST for varying $m$, the number of noise samples used for estimating smoothed predictions. We assume $\sigma = 1.0$ in this experiment. "Gaussian" indicates the baseline training with Gaussian augmentation. We set the results bold-faced whenever the value improves "Gaussian".

| Setups | ACR | 0.00 | 0.25 | 0.50 | 0.75 | 1.00 | 1.25 | 1.50 | 1.75 | 2.00 | 2.25 | 2.50 |
|---|---|---|---|---|---|---|---|---|---|---|---|---|
| Gaussian | 1.620 | 96.4 | 94.4 | 91.4 | 87.0 | 79.9 | 71.0 | 59.6 | 46.2 | 32.6 | 19.7 | 10.8 |
| $m = 1$ | **1.744** | 94.5 | 92.2 | 88.9 | 84.1 | 78.1 | 70.9 | **61.9** | **51.7** | **41.7** | **31.9** | **23.2** |
| $m = 2$ | **1.776** | 95.3 | 93.0 | 89.8 | 85.4 | 79.8 | **72.7** | **63.5** | **53.1** | **42.6** | **33.0** | **24.0** |
| $m = 4$ | **1.789** | 95.5 | 93.6 | 90.5 | 86.2 | **80.7** | **73.7** | **64.1** | **53.9** | **43.1** | **33.5** | **24.1** |
| $m = 8$ | **1.788** | 95.9 | 93.9 | 91.0 | 86.7 | **81.0** | **73.9** | **64.6** | **54.1** | **43.2** | **33.1** | **23.3** |

# I. Discussion on input-dependent designs of noise scales

In this section, we show that if one allows *different* noise scales for each input in attempt to generalize the current framework of randomized smoothing (Cohen et al., 2019), then the actual robustness guarantee would rapidly decrease as the input dimension grows. In particular, we consider the following classifier $\tilde{f}$ generalizing (2) with some non-negative function $g : \mathbb{R}^d \to \mathbb{R}_{\geq 0}$, defined as follows:

$$\tilde{f}(x) := \arg\max_{c \in \mathcal{Y}} \mathbb{P}_{\delta \sim \mathcal{N}(0, g(x)I)}(f(x + \delta) = c),$$

In other words, we assume that the scaling parameter of the smoothing noise can now be a function of $x$. As in the main text, we are interested in the certified radius $\underline{R}(\tilde{f}; x, y)$ of $\tilde{f}$.

One may expect that $\underline{R}(\tilde{f}; x, y)$ can be significantly larger than $\underline{R}(\hat{f}; x, y)$ since $\hat{f}$ is a special case of $\tilde{f}$, *i.e.*, constant $g(x)$. However, we show that it may not be true for high-dimensional inputs: even a small deviation of $g(x)$ can incur very poor certified robustness. Formally, we prove the following theorem.

**Theorem 1.** *Let $r_i, i \in \mathbb{N}$ be any i.i.d. random variables of zero mean, unit variance, and $\mathbb{E}[r_i^4] < \infty$. Let $\mathcal{F}_d$ be a collection of all measurable functions from $\mathbb{R}^d$ to $\{0, 1\}$. Let $p \in (0.5, 1)$, $\sigma, \tau > 0$, and $\varepsilon \in (0, 1/2]$ be constants such that $\sigma \neq \tau$. Then, for $\delta := (r_1, \ldots, r_d)$, for any $c \in \{0, 1\}$, and for any $d \in \mathbb{N}$, the following statements hold:*

$$\sup_{x, x' \in \mathbb{R}^d : \|x - x'\|_2 \leq \varepsilon} \inf_{f \in \mathcal{F}_d : \mathbb{P}(f(x + \sigma\delta) = c) = p} \mathbb{P}(f(x' + \tau\delta) = c) \leq C/d.$$

*for some constant $C > 0$ which is a function of other constants $p, \sigma, \tau, \varepsilon, \mathbb{E}[r_i^4]$.*

Theorem 1 indicates the curse of dimensionality for the worst classifier under general noises of a finite kurtosis. In particular, it states that there exists an upper bound on $\mathbb{P}(f(x + \tau\delta) = c)$ inversely proportional to the input dimension $d$ even though two inputs $x, x'$ are extremely close. Hence, if we utilize different noise scales (*i.e.*, $\sigma$) for each input, the resulting lower bound on the certified radius relying on the worst-case bound as in (Cohen et al., 2019; Lecuyer et al., 2019; Salman et al., 2019) will be small for high-dimensional inputs. Namely, choosing (almost) constant noise scale for the inputs in the target certification region is necessary.

## I.1. Proof of Theorem 1

We first define $z = (z_1, \ldots, z_d) := x' - x$, *i.e.*, $\|z\|_2 \leq \varepsilon$. Then, the following inequality trivially holds.

$$\inf_{f \in \mathcal{F}_d : \mathbb{E}[f(x + \sigma\delta)] = p} \mathbb{E}[f(x' + \tau\delta)] \leq \inf_{\mathcal{U} \subset \mathbb{R}^d : \mathbb{P}(\sigma\delta \in \mathcal{U}) = p} \mathbb{P}(\tau\delta + z \in \mathcal{U})$$

$$\leq \mathbb{P}\left(\frac{\|\tau\delta + z\|_2^2}{d} \in [\sigma^2 - k, \sigma^2 + k]\right) \tag{13}$$

where $k$ is a non-negative number satisfying

$$\mathbb{P}\left(\frac{\|\sigma\delta\|_2^2}{d} \in [\sigma^2 - k, \sigma^2 + k]\right) = p.$$

The following lemma asserts that the RHS of (13) is bounded by $C/d$ where $C$ is some constant which is only a function of $\mathbb{E}[r_i^4], \sigma, \tau, \varepsilon, p$. This completes the proof of Theorem 1.

**Lemma 2.** *There exists $C$ which is a function of $\mathbb{E}[r_i^4], \sigma, \tau, \varepsilon, p$ such that the following statements hold: for any $d \in \mathbb{N}$ and for any $z \in \mathbb{R}^d$ satisfying $\|z\|_2 \leq \varepsilon$,*

$$\mathbb{P}\left(\frac{\|\tau\delta + z\|_2^2}{d} \in [\sigma^2 - k, \sigma^2 + k]\right) \leq \frac{C}{d}.$$

## I.2. Proof of Lemma 2

Lemma 2 is a direct consequence of the law of large numbers applied to the i.i.d. random variables $r_i^2$. First, we compute the variance of $\frac{\|\sigma\delta\|_2^2}{d}$ using the following equality: for $\eta := \sqrt{\mathbb{E}[r_i^4] - 1}$,

$$
\begin{aligned}
\mathrm{Var}\left(\frac{\|\sigma\delta\|_2^2}{d}\right) &= \mathbb{E}\left[\left(\frac{\|\sigma\delta\|_2^2}{d} - \sigma^2\right)^2\right] = \mathbb{E}\left[\left(\frac{\sigma^2}{d}\sum_{i=1}^{d}(r_i^2 - 1)\right)^2\right] \\
&= \frac{\sigma^4}{d^2}\sum_{i=1}^{d}\mathbb{E}\left[(r_i^2 - 1)^2\right] = \frac{\sigma^4}{d^2}\sum_{i=1}^{d}\mathbb{E}[r_i^4] - 1 \\
&= \frac{\sigma^4(\mathbb{E}[r_i^4] - 1)}{d} = \frac{\sigma^4\eta^2}{d}
\end{aligned}
$$

where the third equality follows from the independence of $r_i$s and the fourth inequality follows from $\mathbb{E}[r_i^2] = 1$. Hence, from the Chebyshev's inequality, we have

$$
\mathbb{P}\left(\left|\frac{\|\sigma\delta\|_2^2}{d} - \sigma^2\right| < \frac{\sigma^2\eta}{\sqrt{d(1-p)}}\right) \geq 1 - (\sqrt{1-p})^2 = p, \tag{14}
$$

i.e., $k \leq \frac{\sigma^2\eta}{\sqrt{d(1-p)}}$.

Now, we derive a similar concentration inequality for $\frac{\|\tau\delta+z\|_2^2}{d}$. To this end, we bound its deviation from $\tau^2 + \frac{\|z\|_2^2}{d}$ as follows:

$$
\begin{aligned}
&\mathbb{P}\left(\left|\frac{\|\tau\delta + z\|_2^2}{d} - \left(\tau^2 + \frac{\|z\|_2^2}{d}\right)\right| \geq \frac{\sigma^2 + \tau^2}{3}\right) \\
&= 1 - \mathbb{P}\left(\left|\frac{\|\tau\delta + z\|_2^2}{d} - \left(\tau^2 + \frac{\|z\|_2^2}{d}\right)\right| < \frac{\sigma^2 + \tau^2}{3}\right) \\
&\leq 1 - \mathbb{P}\left(\left|\frac{\|\tau\delta\|_2^2}{d} - \tau^2\right| < \frac{\sigma^2 + \tau^2}{6} \quad \text{and} \quad \left|\frac{2\tau\sum_{i=1}^{d}r_i z_i}{d}\right| < \frac{\sigma^2 + \tau^2}{6}\right) \\
&= \mathbb{P}\left(\left|\frac{\|\tau\delta\|_2^2}{d} - \tau^2\right| \geq \frac{\sigma^2 + \tau^2}{6} \quad \text{or} \quad \left|\frac{2\tau\sum_{i=1}^{d}r_i z_i}{d}\right| \geq \frac{\sigma^2 + \tau^2}{6}\right) \\
&\leq \mathbb{P}\left(\left|\frac{\|\tau\delta\|_2^2}{d} - \tau^2\right| \geq \frac{\sigma^2 + \tau^2}{6}\right) + \mathbb{P}\left(\left|\frac{2\tau\sum_{i=1}^{d}r_i z_i}{d}\right| \geq \frac{\sigma^2 + \tau^2}{6}\right) \\
&\leq \frac{36\tau^4\eta^2 + 144\tau^2\varepsilon^2}{(\sigma^2 + \tau^2)^2 d}
\end{aligned} \tag{15}
$$

where the last inequality is from the variance bounds

$$
\mathrm{Var}\left(\frac{\|\tau\delta\|_2^2}{d} - \tau^2\right) = \frac{\tau^4\eta^2}{d}
$$

$$
\mathrm{Var}\left(\frac{2\tau\sum_{i=1}^{d}r_i z_i}{d}\right) = \frac{4\tau^2\|z\|_2^2}{d^2} \leq \frac{4\tau^2\varepsilon^2}{d^2}
$$

and the Chebyshev's inequality

$$
\mathbb{P}\left(\left|\frac{\|\tau\delta\|_2^2}{d} - \tau^2\right| \geq \frac{\sigma^2 + \tau^2}{6}\right) \leq \frac{36\tau^4\eta^2}{(\sigma^2 + \tau^2)^2 d}
$$

$$
\mathbb{P}\left(\left|\frac{2\tau\sum_{i=1}^{d}r_i z_i}{d}\right| \geq \frac{\sigma^2 + \tau^2}{6}\right) \leq \frac{144\tau^2\varepsilon^2}{(\sigma^2 + \tau^2)^2 d^2}.
$$

Then, for all $d \geq \max\left\{\frac{4\sigma^4\eta^2}{(\tau^2-\sigma^2)^2(1-p)}, \frac{6\varepsilon^2}{\sigma^2+\tau^2}\right\}$, *i.e.*, $\frac{\sigma^2\eta}{\sqrt{d(1-p)}} \leq \frac{|\tau^2-\sigma^2|}{2}$ and $\frac{\varepsilon^2}{d} \leq \frac{\sigma^2+\tau^2}{6}$, it holds that

$$\mathbb{P}\left(\frac{\|\tau\delta+z\|_2^2}{d} \in [\sigma^2-k, \sigma^2+k]\right)$$

$$\leq \mathbb{P}\left(\frac{\|\tau\delta+z\|_2^2}{d} \in \left[\sigma^2-\frac{|\tau^2-\sigma^2|}{2}, \sigma^2+\frac{|\tau^2-\sigma^2|}{2}\right]\right)$$

$$\leq \mathbb{P}\left(\left|\frac{\|\tau\delta+z\|_2^2}{d}-\left(\tau^2+\frac{\|z\|_2^2}{d}\right)\right| \geq \frac{\sigma^2+\tau^2}{3}\right)$$

$$\leq \frac{36\tau^4\eta^2+144\tau^2\varepsilon^2}{(\sigma^2+\tau^2)^2 d}$$

by using (15). Hence, choosing

$$C := \max\left\{\frac{4\sigma^4\eta^2}{(\tau^2-\sigma^2)^2(1-p)}, \frac{6\varepsilon^2}{\sigma^2+\tau^2}, \frac{36\tau^4\eta^2+144\tau^2\varepsilon^2}{(\sigma^2+\tau^2)^2}\right\}$$

completes the proof of Lemma 2.