# OpenReview forum: "SmoothMix: Training Confidence-calibrated Smoothed Classifiers for Certified Adversarial Robustness"
_ICML.cc/2021/Workshop/AML — ICML 2021 Workshop AML Poster_

### Official Review · Reviewer_xSez · 2021-06-19
**A novel adversarial training scheme for improving certified robustness**

**Rating:** Accept
**Confidence:** 4

**Review:**

The authors proposed SmoothnMix which trains neural networks by using convex combinations of adversarial perturbations.  By adding a mixup loss as a regularization term, the confidence of over-confident samples can be calibrated. The experimental results show that there is a small but effective improvement by adding this regularization term.

A typo in line 24, "the smoothed classifiers offer is a direct ..."

---

### Decision · Program_Chairs · 2021-06-21

**Decision:**

Accept (Poster)

**Comment:**

This paper proposed SmoothnMix which trains neural networks by using convex combinations of adversarial perturbations. The paper is suitable for the workshop.